# Exploring African Community Attitudes Towards Mental Illness in Australia: A Cross-Sectional Study

**DOI:** 10.3390/healthcare13233115

**Published:** 2025-12-01

**Authors:** Gihane Endrawes, Olayide Ogunsiji

**Affiliations:** School of Nursing and Midwifery, Western Sydney University, Penrith, NSW 2116, Australia; o.ogunsiji@westernsydney.edu.au

**Keywords:** African communities, attitudes to mental illness, mental illness, stigma, psychometric properties

## Abstract

Background: Mental illness is often stigmatized across various cultural groups, yet there is limited understanding of African communities’ perceptions and beliefs regarding mental health. One reason for this disparity could be the lack of culturally appropriate tools to assess attitudes towards mental illness in African populations. This study aimed to evaluate the psychometric properties of the 40-item Community Attitudes towards the Mentally Ill (CAMI) scale within African communities in Australia. Design: This study employed a quantitative, cross-sectional, and descriptive approach, using a self-administered survey to assess the psychometric properties of the CAMI scale among African Australian individuals. Cronbach’s alpha was used to evaluate internal consistency. Method: A convenience sample of 110 individuals from various African community organizations in Australia was recruited. The original English version of the CAMI scale was used to assess attitudes towards mental illness. Results: Cronbach’s alpha for the overall scale was 0.717, indicating acceptable consistency. The Authoritarianism sub-scale had a lower reliability of 0.424, which is below the acceptable threshold of 0.70. The other sub-scales had a better internal consistency, with 0.730 for Benevolence, 0.724 for Ideology, and 0.627 for Social Restrictiveness, though the latter still lacked the ideal 0.70. Conclusions: The CAMI scale has been demonstrated to be a reliable and culturally appropriate tool for assessing African communities’ attitudes towards mental illness in Australia. By identifying negative attitudes, this tool can be used to inform health education and awareness programs that address misconceptions about mental illness. Such programs could encourage early health-seeking behaviors among migrants, facilitating early identification and intervention, and ultimately improving health outcomes by reducing the burden of mental illness. This study is significant as it provides a culturally appropriate tool to assess mental health attitudes in African communities, informing the development of appropriate strategies to promote early help-seeking behaviors and reduce stigma, thus improving mental health outcomes.

## 1. Introduction

Mental illness is a widespread issue affecting a significant portion of the global population, with approximately one in five adults worldwide experiencing a diagnosable mental health condition annually [1]. Among refugees and asylum seekers, particularly those arriving through humanitarian programs, mental health struggles are more prevalent due to war, displacement, and trauma-related symptoms such as PTSD and depression [2,3]. Resettled refugees often face additional challenges like anxiety and depression linked to social integration difficulties, yet cultural stigma discourages many from accessing mental health services [4].

Australia is one of the most culturally diverse nations globally, with over half the population either born overseas or having at least one parent born abroad. African migrants, estimated at 496,000, represent about 1.7% of the Australian population and 5.1% of those born overseas [5]. Despite their growing visibility, research on African migrants’ mental health perspectives remains limited. There is an urgent need to understand their attitudes toward mental illness, especially given the cultural and systemic barriers to care [6]. Mental health issues are prevalent across the African communities, with approximately 27% of adolescents experiencing psychiatric distress [7]. Disorders such as anxiety, depression, and suicidal tendencies are increasingly common, yet stigma and misperceptions often prevent families from seeking professional help [8]. In many African and similar communities, mental illness is viewed negatively and attributed to weakness, evil spirit possession and other spiritual causes, resulting in families seeking help from traditional and spiritual healers rather than health care services [9,10]. The lack of culturally sensitive care also contributes to the delayed access to mental health services [11,12,13]. Developing culturally appropriate mental health services tailored to these populations is essential [10].

Given the prevalence of mental health issues and stigma among migrants, there is a pressing need for targeted research exploring attitudes within Australia’s African migrant communities [14]. Several tools have explored stigma and attitudes toward mental illness, including the Custodial Mental Illness Ideology (CMI) Scale [15], the Opinion about Mental Illness (OMI) Scale [12], and the Mental Health Knowledge Schedule (MAKS) [16]. This study examines attitudes toward mental illness within African migrants in Australia using the Community Attitudes towards the Mentally Ill (CAMI) scale. The CAMI scale was originally developed and validated by [17] in Toronto/Canada. Three out of the four sub-scales had high reliability, ranging from 0.88 for social restrictiveness to 0.80, and benevolence at 0.76. The coefficient for authoritarianism was lower (0.68), but still satisfactory. To validate the scale cross-culturally, it has been translated into different languages such as Chinese [18], French [19] and Swedish [20]. Ref. [21] used CAMI-31 items with the general population in the United Kingdom, while ref. [22] used CAMI-W 20 items among Asian, Caribbean, and African communities living in London. Ref. [19] used CAMI-26 items in France with undergraduate nursing students. These studies reported that Cronbach’s alpha values were below the minimum cut-off point recommended in the original CAMI sub-scale, but did not explicitly report on their values. The study examined the construct validity of the modified 20-item CAMI scale among 421 Swedish nursing students using principal component analysis and showed moderate–high loading for the 20-item scale but not for the original 40-item scale [20]. Cronbach’s alpha scores for the 3 extracted factors were 0·84, 0·77, and 0·71, validating the appropriateness of the modified CAMI scale among this cohort. Few studies assessed the psychometric properties of the original 40-item CAMI scale, and results showed variations in the scale’s structure, especially when used cross-culturally. For example, a validation study among the Taiwanese population revealed a 4-factor model with different item loads compared to the original CAMI scale [23]. Similarly, a UK study [21] identified a 3-factor model instead of the original 4-factor structure. More recently, ref. [24] examined measurement invariance of stigma scales across Chinese and US samples; ref. [25] conducted cultural adaptation and validation of a stigma scale for Arabic-speaking populations, and refs. [25,26,27] demonstrated rigorous adaptation and validation processes in Latin American and African contexts.

For this study, the CAMI scale was chosen as it is considered the ‘gold standard’ for examining stigma associated with mental illness and has been validated across diverse cultural contexts [28]. This research provides preliminary findings of a study providing a contextually reliable tool for the measurement of attitudes towards mental illness. It addresses a significant gap by exploring African migrants’ attitudes towards mental illness using the adapted CAMI scale. Understanding these attitudes is essential for developing effective public health strategies that reduce stigma, promote early help-seeking, and improve mental health outcomes for African migrants and refugees.

## 2. Methods

### 2.1. Study Design

This research employed a cross-sectional descriptive design involving a self-administered survey.

### 2.2. Sampling and Data Collection

The target population comprised individuals from African backgrounds living in Australia. In this study, “African communities” refers to individuals of African heritage who have migrated to Australia, including both refugees and non-refugee migrants. Participants self-identified as being of African background and were born overseas. This group was selected due to the growing African migrant population in Australia and the limited research exploring their culturally influenced attitudes toward mental illness. We obtained ethics approval from the Institutional Human Ethics Committee. Our approach to recruitment of participants was informed by previous studies among African migrants, and what worked in accessing them in Australia [6] and other international countries. For this study, we utilized a convenience sampling technique to recruit participants from religious organizations, African community events, and end-of-the-year activities, as well as through personal networks. A group of 110 African Australians was recruited through various African community organizations, including churches and social gatherings. The inclusion criteria required participants to be aged 18 years or older; (2) self-identifying as being of African backgrounds and born overseas; and (3) capable of reading and writing in English. The exclusion criteria included individuals who did not identify as Africans, individuals who did not read or write in English, and individuals who are younger than 18 years old.

Before data collection, information sheets containing details of the study were distributed to the members of the religious and community groups. The second author provided detailed information about the study, the questionnaire method of data collection, associated risks and benefits of participation, and the need for consent. All the potential participants had opportunities to ask questions and clarify their thoughts. It was clearly explained to them that participation was voluntary and that they could withdraw their participation at any time without negative implications on their membership in the organizations. Participants who provided consent to participate in the study were asked to complete the CAMI scale, which took approximately 12–20 min. Additional explanation and time were spent with participants with a lower level of English language proficiency. Arrangement was made with the leaders of the organizations, by the second author, for collection of the completed questionnaire.

### 2.3. Sample Size

There is no universally accepted sample size for reliable factor analysis and psychometric validation. However, a common recommendation is to have 5–10 participants per item on the scale. Given that the Community Attitudes Towards the Mentally Ill (CAMI) Scale contains 40 items, a sample size ranging from 200 to 400 participants is ideal for validation [29]. In this study, 200 questionnaires were distributed to potential participants, and 110 responses were returned, resulting in a 55% response rate. Although this is a modest response rate, it is deemed adequate, particularly given the sensitivity of the topic of mental illness, which is highly stigmatized among the African community [30]. In practice, many published validation studies use sample sizes between 100 and 200, particularly when working with hard-to-reach populations, such as African populations in this case [30,31,32].

### 2.4. Instrument

CAMI Scale Adaptation and Validation

Ref. [19] developed the Community Attitudes Towards the Mentally Ill (CAMI) scale in Toronto, Canada, and validated it for use in multiple settings. The original CAMI scale contains 40 items rated on a five-point Likert scale (1—strongly agree, 2—agree, 3—unsure, 4—disagree, and 5—strongly disagree) and is divided into four subscales: (a) Authoritarianism (AU)—reflects views of people with mental illness as inferior and in need of coercive treatment, (b) Benevolence (BE)—captures paternalistic and sympathetic attitudes toward people with mental illness, (c) Social Restrictiveness (SR)—concerns the belief that people with mental illness pose a threat to society and should be avoided, (d) Community Mental Health Ideology (CMHI)—involves the acceptance of mental health services and individuals with mental illness in the community. The CAMI scale was piloted with eight African Australian participants to ensure face validity, focusing on clarity, understanding, and readability. No issues with problematic language or unclear items were identified, and the scale was determined to be culturally relevant and appropriate for assessing social stigma and attitudes toward mental illness in African communities in Australia.

In this study, two items were intentionally excluded from analysis due to conceptual redundancy: “The mentally ill should not be treated as outcasts of society” (Authoritarianism) and “There are sufficient existing services for the mentally ill” (Benevolence). The first item overlaps in meaning with other items promoting inclusion and responsibility, such as “Mental health patients should be encouraged to assume the responsibilities of normal life.” The second item reflects systemic attitudes toward service adequacy, which are already addressed in items like “Mental hospitals are an outdated means of treating the mentally ill” and “As far as possible, mental health services should be provided through community-based facilities.” Excluding these items helped reduce redundancy and improve the psychometric clarity of the subscales. This approach is consistent with practices in CAMI validation and adaptation studies. For example, ref. [28] conducted a systematic review of CAMI applications and highlighted that item reduction is common, particularly in the Authoritarianism subscale, which often shows low internal consistency. Similarly, refs. [33,34] noted that several studies have excluded CAMI items due to poor psychometric performance or lack of conceptual clarity, which highlights the need for item refinement in CAMI applications to enhance reliability and construct validity.

### 2.5. Data Analysis

The CAMI scale was rated on a five-point scale, where a higher score indicates a more open-minded approach to individuals with mental illness. For all analyses, the scores were re-arranged, if necessary, so that higher values corresponded to more favorable attitudes towards mental health for all items. Descriptive statistics were used to describe the demographic characteristics of the study participants. Cronbach’s alpha was calculated for each of the four domains of the CAMI scale to evaluate internal consistency. A Cronbach’s alpha value of 0.7 or above is considered to indicate a good internal consistency, while values above 0.9 may indicate redundancy in items [34,35,36].

The Cronbach’s alpha for each subscale was determined, along with the correlation of each subscale with other subscales and the total score. Additionally, mean subscale and total scores were calculated and compared across subgroups based on factors such as language version used, gender, religion, whether participants had children, and whether they knew someone with mental illness.

The data analysis in this study utilized two key methods to evaluate the psychometric properties of the CAMI scale. Initially, reliability testing was conducted using Cronbach’s Alpha to determine the internal consistency of the CAMI scale and its subscales. This measure indicates how consistently the items on the scale reflect the same concept, with values of 0.7 or higher typically indicating satisfactory reliability. In addition to Cronbach’s Alpha, split-half reliability was utilized, dividing the dataset into two halves and correlating the scores from each, with adjustments made using the Spearman–Brown formula to determine the number of items.

Second, the study employed Pearson’s Product–Moment Correlation to examine the relationships between the four subscales of the CAMI scale (Authoritarianism, Benevolence, Social Restrictiveness, and Mental Health Ideology) and the overall Community Attitude Toward Mental Illness (CAMI) score. The correlation coefficients provided insights into the direction and strength of the relationships between these components, enhancing understanding of how various attitudes toward mental illness are interrelated within the African migrant population in Australia. The combination of reliability analysis and correlation led to a thorough examination of the scale’s psychometric properties and its applicability in this context.

## 3. Results

A total of 110 African Australian individuals agreed to participate in the study, and their demographics are summarized (see Table 1). In terms of gender, 59 (53.6%) identified as male, while 51 (46.4%) identified as female. For age, 50 (45.5%) were between 20 and 30 years old, 17 (15.5%) were in the 31–40 age range, 5 (4.5%) were aged 41–50, 3 (2.7%) were aged 51–60, and 35 (31.8%) did not provide a response. Regarding religious affiliation, 90 (81.8%) were Christians, 16 (14.5%) were Muslims, and 4 (3.6%) did not specify their religion. The majority of respondents, 102 (92.7%), were from West Africa, with 1 (0.9%) from East Africa, 1 (0.9%) from North Africa, and 6 (5.5%) did not answer. Nationality-wise, 44 (40.0%) were from Nigeria, 3 (2.7%) were from Kenya, and 63 (57.3%) did not disclose their nationality. Regarding marital status, 43 (39.1%) were married, 64 (58.2%) were single, 1 (0.9%) was divorced, and 2 (1.8%) did not respond. Concerning children, 39 (35.5%) of the respondents had children, while 66 (60.0%) did not, and 5 (4.5%) did not answer this question. When asked about the duration of their stay in Australia, 5 (4.5%) had lived there for 1–20 years, 6 (5.5%) for 21–30 years, 2 (1.8%) for 31–40 years, and a significant 97 (88.2%) did not answer. As for educational background, 5 (4.5%) never attended school, 2 (1.8%) had a primary school certificate, 4 (3.6%) had a secondary school certificate, 3 (2.7%) had a TAFE/college certificate, 85 (77.3%) held a tertiary degree, and 11 (10.0%) did not respond. For English proficiency, 65 (59.1%) rated their spoken English as very good, 36 (32.7%) as good, 4 (3.6%) as average, and 1 (0.9%) rated their spoken English as limited. Employment status showed that 46 (41.8%) were unemployed, 54 (49.1%) were employed, 1 (0.9%) was retired, and 9 (8.2%) did not provide information.

All surveys, including those missing some demographics, were included as the main aim of the study was to examine African migrants’ attitudes toward people with mental illness collectively, rather than examining the impact of various demographics on attitudes.

Appendix A delineates notable instances of non-responses for specific demographic variables, while the CAMI items demonstrated a minimal rate of missing responses. Importantly, all 110 participants completed the 38 CAMI items, with none exhibiting missing values that exceeded the threshold for imputation via the half-rule. Consequently, no CAMI items necessitated imputation, nor did any participants require imputation for CAMI data. This clarification has been incorporated into the revised manuscript.

The half-rule was not applied to demographic variables, as they represent individual data points rather than scale items. Analyses involving demographic data were conducted using available case data only, in accordance with recommended practices for managing missing data in survey research [33,34].

Table 1 presents responses to statements indicating authoritarian attitudes toward mental illness. The data provides a range of opinions on mental health issues, with varying levels of agreement or disagreement. For instance, 45.4% of respondents disagreed with the assertion that one of the primary causes of mental illness is a lack of self-discipline and willpower, indicating a substantial portion of individuals refused this view. Similarly, a majority of 70% disagreed with the idea that the best way to handle mentally ill individuals is to keep them behind locked doors, suggesting that respondents preferred less restrictive approaches to mental health care. Nonetheless, a large number of individuals, 72.7%, agreed that there is something about mentally ill individuals that makes them easily distinguishable from those without mental health issues, indicating a more stigmatized view of mental illness.

Further analysis reveals mixed opinions on the treatment of individuals with lived experience. According to 69.1% of respondents, individuals with mental disturbance should be hospitalized immediately, and 67.3% agreed that mental health patients need the same kind of control and discipline as young children. These viewpoints reflect a desire for greater control and structure in mental health treatment. In contrast, 55.4% of respondents disagreed with the notion that less emphasis should be placed on protecting the public from mentally ill individuals, indicating ongoing safety concerns. Additionally, 76.4% of respondents stated that anyone can become mentally ill, highlighting a more empathetic understanding of mental illness. The data suggest a combination of authoritarian and more progressive attitudes towards mental health, with a tendency towards the need for structure and protection, but also a recognition of the universality of mental health issues.

Table 2 presents responses to statements expressing benevolent attitudes toward the mentally ill. The majority of respondents expressed their support for more compassionate views, with 84.7% stating that the mentally ill have been the subject of ridicule, and 85.5% advocating for increased tax funding for the care and treatment of the mentally ill. A strong 89.1% believed that society should adopt a more tolerant attitude towards individuals with mental health issues. When it comes to mental health care settings, 76.4% of respondents stated that mental hospitals often resemble prisons rather than places of care, indicating concerns about the treatment area. Additionally, 87.3% believed that society must provide the best possible care for the mentally ill. However, there were contrasting viewpoints, as 54.5% disagreed with the notion that the mentally ill deserve sympathy, and 60% felt that the mentally ill were not a burden on society. Furthermore, 81.8% rejected the idea that enlarged spending on mental health services is a waste of taxpayer money, while 68.2% disagreed with avoiding persons with mental health problems, showing that, notwithstanding some negative perceptions, a large proportion of respondents supported greater empathy and investment in mental health care.

Table 3 presents answers to statements regarding social restrictiveness towards individuals with mental illness. The data shows a mixture of negative and more progressive views. For example, a considerable portion of respondents, 42.8%, disagreed with the idea that the mentally ill should not be given any responsibility, while 42.7% disagreed with isolating the mentally ill from the rest of the community. Furthermore, 71.8% refused the notion that a person would be crazy to marry someone with a history of mental illness, and 58.2% expressed no objection to living next door to someone who has been mentally ill. On the other hand, the idea of disregarding individuals with mental illness from public office had mixed responses, with 40.9% disagreeing, while 25.5% supported exclusion. Additionally, the majority of respondents, 82.7%, agreed that the mentally ill should not be denied their individual rights, and 71.8% agreed that mental health patients should be encouraged to assume the responsibilities of normal life. However, there were some more restrictive views, with 47.3% of respondents believing that mental health patients can be trusted as babysitters, indicating a significant portion of individuals still held some level of skepticism about the ability of the mentally ill to fully integrate into society. Overall, the responses reflect a mix of progressive opinions and persistent social restrictiveness towards individuals with mental health conditions.

Table 4 provides insight into public attitudes toward the location of mental health facilities in residential neighborhoods. A majority of respondents, 68.2%, agreed that residents should accept the placement of mental health facilities in their neighborhoods, and 66.4% believed that being part of a normal community is beneficial therapy for many mental health patients. Furthermore, 80.9% of respondents supported the idea of providing mental health services through community-based facilities. However, concerns about safety were evident, as 56.3% agreed that locating mental health services in residential areas does not pose a threat to local residents, and 69.0% felt that residents have nothing to fear from people accessing mental health services in their neighborhood.

On the other hand, opinions were more divided when it came to the potential risks of having mental health facilities nearby. While 45.5% disagreed with the idea that mental health facilities should be excluded from residential areas, 56.4% felt that local residents had valid reasons to resist such locations. A significant portion, 65.4%, thought that while mental health patients living in residential neighborhoods might benefit from therapy, the risks for residents were too high. Additionally, half of the respondents (50.9%) found the idea of people with mental health problems living nearby frightening, while 61.8% disagreed with the statement that locating mental health facilities would downgrade the neighborhood. These responses display a complex balance between support for community-based mental health care and concerns about the potential risks and challenges it may bring to residential areas.

Table 5 presents the reliability analysis of the scales used in the study, including Cronbach’s Alpha, Split-Half Reliability Coefficient, and correlations between forms. Item analysis helps determine whether or not an item is contributing to the internal consistency of the scale. The internal consistency of a scale, in turn, influences the reliability coefficient of the overall scale. Ref. [31] argued for the use of scales with modest reliability coefficients to make sound decisions. The authors proposed some guidelines on how to interpret the reliability coefficients. Reliability coefficient (Cronbach’s alpha) values above 0.70 and item-total correlational values above 0.20 were considered acceptable in this study [36,37]. The Cronbach’s alpha values for the scales range from 0.424 for Authoritarianism to 0.730 for Benevolence, with Benevolence, Ideology, and Community Attitude Towards Mental Illness meeting the standard reliability threshold of 0.70. However, Authoritarianism and Social Restrictiveness fall below this threshold (α = 0.42) and (α = 0.62), respectively, indicating lower internal consistency. These lower reliability scores may reflect cultural differences in how these constructs are understood within African migrant communities in Australia. These values are consistent with earlier reports, such as the original CAMI validation by [19], which also found lower reliability for these subscales. While the overall scale shows acceptable reliability, these subscales may require further refinement to better capture the attitudes of this population.

The split-half reliability coefficient assesses the internal consistency of a scale by correlating scores from two halves of the test. Values closer to 1 indicate higher reliability. In psychometric research, coefficients above 0.70 are generally considered acceptable, above 0.80 good, and above 0.90 excellent [35,36].

The Spearman–Brown coefficient adjusts the split-half correlation to estimate the reliability of the full-length scale, compensating for the reduction in reliability caused by halving the test [37]. Its interpretation follows the same thresholds as above. A high Spearman–Brown coefficient (e.g., ≥0.80) indicates that the scale has good internal consistency and is suitable for research or clinical use [37].

There was a significant relationship among authoritarian (r = 0.52, *p* < 0.01), benevolence (r = 0.41, *p* < 0.01), social restrictiveness (r = 0.22, *p* < 0.05), mental health ideology (r = 0.46, *p* < 0.01) and community attitude towards mental illness (see Appendix A). This result shows that an increase in authoritarianism, benevolence, social restrictiveness and mental health ideology was associated with a significant increase in Positive community attitudes towards mental illness. People who scored higher on Benevolence (showing more empathy and kindness) usually scored lower on Social Restrictiveness (showing less support for isolating or limiting freedoms of people with mental illness). Similarly, those who supported Community Mental Health Ideology (supporting the integration of people with mental illness into the community) also showed lower Social Restrictiveness. These negative links are expected because they show opposite relationships between good and bad attitudes.

## 4. Discussion

According to the knowledge of authors at the time of publication, this is the first investigation into African community attitudes toward people with mental illness in Australia, utilizing the 40-item African CAMI scale. The study examined the psychometric reliability of the CAMI scale for African migrant populations. Although the CAMI scale is widely used in mental health research, its applicability and reliability in assessing attitudes towards mental illness among culturally diverse individuals have not been extensively studied [38]. This research contributes to the broader field of cross-cultural research in mental health Insights gained from the CAMI scale’s sub-scales, particularly the Authoritarianism sub-scale, provide valuable information on how stigma exists within the African migrant population.

In line with cross-cultural research, the majority of respondents (45.5%) were aged between 20 and 30 years, suggesting that younger individuals are more engaged in such research. This highlights the need to address the reasons behind older African migrants’ low engagement with research and, in particular, mental health research. Ref. [39] pointed out that some of the reasons are culturally based and issues of stigma surrounding mental illness among African communities, as well as a lack of trust in people outside the family, are of importance. There is a need to explore best practices that are evidence-based to increase older migrants’ participation in research. This is important to ensure adequate representation and inclusivity of all age groups and to get the voices of all age groups heard. Thus, identifying specific age-related barriers to mental health service utilization requires urgent attention. The outcome of such research will inform future directions, research, policies, and strategies targeting the needs of the diverse African communities and addressing recruitment issues.

Apart from the role of age with participation in mental health research, age also influences health-seeking behaviors for mental health issues. Consistent with international studies [40,41], younger generations are more accepting of mental illness and more likely to seek professional help for their mental health issues compared to older generations. This could be related to stigmatizing attitudes of the older generation, who might have migrated to Australia at an older age with a lower acculturation level, including attitudes toward mental illness, compared to younger generations who would have received their education and received mental health literacy during schooling [8,39].

### 4.1. Acculturation Variable Impact on Attitudes

In this study, a similar stigmatized view of mental illness was reported among migrants in the United States of America (USA) [42], where a direct correlation was found between acculturation level and stigma. It is important to note that the participants in the USA were international students. Even though the majority of the participants in our current study did not respond to the demographic question on the number of years in Australia, a significant number of had been in Australia for at least 30 years. If one considers that most of these people have tertiary education and that they describe themselves as having good English proficiency, we expect a positive attitude towards mental health. This suggests that regardless of the factors that inform migration or how long one might have been in a new country, a stigmatized perspective of mental illness persists. We suggest targeted messages that address stigma in mental health education among migrant communities.

### 4.2. Financial Investment in Mental Health Services

Insight into the financial implications of mental illness have been previously provided in published literature from the USA [43] and Saudi Arabia [44]. The USA study reported that as high as $774 US dollars go into services such as community health centers, housing and other supports in some of the USA states. Nevertheless, aligning with the findings of our current study, the Saudi Arabian study [44] argued that it is crucial for decision makers such as health officials and policymakers to make substantial investments in mental health and substance use services. This is integral in promotion of prevention and in offering accessible treatment, care, and rehabilitation.

### 4.3. Gender Difference

Though correlates of gender disparity in relation to attitudes towards mental illness among African communities in Australia are not captured in our current study, it is important to report that gender differences exist in attitudes toward mental illness. Previous studies have consistently reported gender disparities in relation to mental illness-associated stigma [45,46,47,48,49]. Refs. [45,50] also reported that female African college students showed fewer stigmatizing attitudes toward mental illness and were more open to disclosing their mental health issues to friends.

In a study of medical learners in Canada, participants who identified as men had more negative attitudes towards self-disclosure of mental illness than those who identified as women [51]. This is similar to medical doctors in Poland, with female doctors reported to have fewer negative attitudes towards the mentally ill than the males [52].

African men reported greater anticipated stigma about disclosing mental illness, which is related to bringing shame and embarrassment to the family, and the cultural role and expectation to view men as strong. This is particularly true in male-dominant contexts, where men are the head of the family [53,54]. We suggest gender-specific interventions for reducing mental-illness-associated stigma among migrant communities and health care providers.

### 4.4. Education Level

In this study, more than half of the participants (56.4%) with the Community Mental Health Ideology felt that local residents had legitimate reasons to resist mental health facilities being located in their neighborhood. This may be due to limited knowledge about the need for timely access to services and may impact their ability to advocate for timely access and proximity to mental health services. A lack of knowledge is identified as a major barrier to mental health service utilization. This aligns with ref. [54] study, showing a positive relationship between having access to mental health knowledge and service utilization. On the other hand, ref. [49] argued that increasing public awareness and knowledge did not necessarily cause a change in attitudes toward mental illness and thus had no impact on service utilization. This could be related to the deeply rooted cultural beliefs and perceptions related to mental illness, which can be difficult to change. Demographic data of our current study reveals that majority of the participants had a tertiary education and were professionals. However, the specifics of their professions are beyond the focus of this study; perhaps they are not health-related. Ref. [14] emphasizes the importance of developing culturally appropriate knowledge-building to reduce stigma and increase help-seeking behavior. African communities in Australia would benefit from community-wide culturally sensitive educational resources on mental health and stigma, regardless of their educational achievement and professional status. This would enhance their capacity to advocate for people living with mental illness in their communities.

### 4.5. Strengths and Limitations

One of the key strengths of this study is to enhance the understanding of mental health professionals regarding African communities’ attitudes toward mental illness. The use of the CAMI scale provided a suitable tool to assess attitudes, which could be beneficial in developing appropriate strategies to address these attitudes. These results can inform the establishment of culturally tailored mental health awareness, education, and promotion programs designed to improve knowledge and attitudes towards mental illness among African Australians.

This study contributes to the field of cross-cultural mental health research, particularly in the context of African migrant and refugee populations in Australia. It provides empirical data on the community’s attitudes towards mental illness within African populations, a topic that has been overlooked in Australian research. While existing studies have documented stigma and negative views toward mental illness in African countries [53,55], there has been limited research on these attitudes within African migrant communities in Western nations such as Australia. This study focuses on African migrant and refugee populations in Australia, providing a deeper understanding of how cultural factors shape attitudes towards mental illness in diaspora communities.

This study provides critical insights for the development of culturally appropriate mental health interventions aimed at addressing the stigma within African migrant communities in Australia. By identifying the specific attitudes and misconceptions that perpetuate stigma, the research outlines important areas of focus in the development of targeted mental health education and awareness programs. These initiatives can reduce stigma, encourage help-seeking behavior, and ultimately lead to better mental health outcomes for African migrants [11,56]. These interventions are essential for promoting social integration, improving mental health, and ensuring migrants receive the support they need. Given these strengths, this study has provided a starting point in gaining deep insight into African migrants’ attitudes towards mental illness in Australia, using a reliable tool, the CAMI scale.

The study has several limitations. A key limitation is that confidence intervals for reliability and correlation estimates were not reported. Confidence intervals are widely recommended to enhance transparency, interpretability, and the practical value of research findings [57,58,59,60]. We recommend that future studies routinely report confidence intervals in line with these best practices.

In terms of sample size, while the ideal range for validating a 40-item instrument is typically between 200 and 400 participants, this study included 110 valid responses. Although smaller than recommended, this sample provides preliminary insights into the instrument’s psychometric properties. We advise interpreting the alpha and correlation coefficients with caution and encourage future research to replicate these findings with larger samples and full reporting of confidence intervals.

A large proportion of participants (81.8%) identified as Christian, which could have influenced attitudes toward mental illness, indicating that the findings may not be generalizable to individuals from other religious beliefs. Additionally, the use of self-reported surveys presents the potential for social desirability bias, where respondents may underreport stigmatized views on mental illness to present themselves in a more favorable environment. Although the CAMI scale was completed without missing responses, demographic data had higher rates of non-response. This could be due to the lack of trust in outsiders, past negative experiences with mental health services, privacy concerns and fear of identification within close-knit African communities in Australia; concerns related to visa status may have discouraged participants from disclosing personal information or unfamiliarity with research methods. Retaining the “no response” category in a survey acknowledges the practical challenges inherent in data collection processes within cross-cultural migrant research. As noted by [8], participants in such studies may hesitate to disclose demographic details due to various personal or cultural reasons. While these missing responses were included in the analysis, the focus of this study remains on understanding attitudes toward mental illness within African communities, rather than emphasizing specific demographic variables. While all surveys were included in the analysis to preserve the integrity of the collective data, the missing demographic information may introduce non-response bias and limit the generalizability of the findings. Future studies should consider strategies to improve response rates for demographic items, such as building trust with participants and reinforcing confidentiality. This highlights the importance of employing a culturally sensitive approach in research involving migrants and culturally diverse populations [4,8]. Future research could examine whether younger individuals were more likely to complete the demographic section than older individuals, as this pattern could have implications for understanding the factors influencing participation.

Another limitation of this study is its cross-sectional design, which limits the psychometric evaluation. We measured internal consistency but did not look at other important factors like test–retest reliability, discriminant validity, and construct validity. This is common in cross-sectional research, where internal consistency is often prioritized over temporal stability [61]. Given our cultural focus, this method was suitable for our goals. However, thorough validation, including repeatability and more validity checks, is important for strong instrument development [61]. These results lay the groundwork for further validation of the CAMI scale in African Australian communities and should be interpreted carefully.

### 4.6. Implications for Research, Practice and Policy

Further validation of the CAMI scale is essential among African populations to assess its applicability across diverse cultural and geographic contexts. While the findings of this study offer valuable insights into attitudes toward mental illness within African Australian communities, they may not be universally generalizable to all African groups. Replicating this research with varied cultural samples would strengthen the scale’s reliability and support meaningful cross-cultural comparisons, enhancing its relevance for international mental health research and practice.

We recommend that future research continue to assess the test–retest reliability of the CAMI scale. Although previous studies have reported mixed results—ranging from acceptable reliability in Chinese and Persian samples to poor reliability in a Kenyan community sample—these findings highlight the importance of evaluating temporal stability across different cultural contexts. Such assessments allow for replication, contextual comparison, and potential refinement of the instrument.

In addition to test–retest reliability, future studies should examine other psychometric properties, including discriminant validity to determine whether CAMI subscales measure distinct constructs, and construct validity through exploratory and confirmatory factor analysis to investigate the underlying structure of the scale. Evaluating the scale’s responsiveness to change is also important, particularly in the context of educational or anti-stigma interventions.

Moreover, future research should address gaps in demographic data and explore the interplay between mental health attitudes and socioeconomic variables. These efforts would contribute to a more comprehensive understanding of the CAMI’s psychometric robustness and its utility in culturally diverse settings, ultimately informing the development of culturally responsive mental health policies and practices.

Mental health nurses and other health care professionals are crucial in dispelling misconceptions, reducing stigma, and promoting awareness regarding mental illness. This is particularly important among African migrant communities, where cultural and religious factors may heavily influence attitudes toward mental illness. The findings of this study have implications for policymakers, indicating the need for appropriate funding and the development of culturally sensitive mental health resources. These resources should be available in a variety of languages and be suitable for all age groups and individuals. Collaboration with community leaders can reduce the stigma and increase knowledge, fostering better mental health outcomes for these communities.

To combat the stigma associated with mental health in African communities in Australia, breaking the silence around mental health is crucial. Educational campaigns can help dispel myths and misconceptions, promoting a more informed understanding of mental health problems. Providing culturally sensitive services is essential for African individuals with mental health concerns. This includes ensuring mental health professionals are trained to respect and comprehend cultural beliefs while providing services that are more accessible and responsive to the specific needs of these communities. Including African community leaders in mental health initiatives can be an effective strategy for reducing stigma, creating safe spaces for open discussions, and fostering a supportive environment. Open discussions across various media channels can help normalize conversations about mental health, reducing the stigma and encouraging individuals to seek help earlier, leading to better health outcomes.

## 5. Conclusions

In Australia, addressing mental health issues in African communities requires a multifaceted and comprehensive approach that incorporates education, culturally sensitive support, and active community involvement. The stigma associated with mental health in African migrant populations is deeply rooted in cultural beliefs, social norms, and religious beliefs, which can lead to barriers in accessing appropriate care. To effectively reduce stigma, it is crucial to foster understanding through culturally relevant educational campaigns that provide accurate information about mental health, challenge harmful stereotypes, and encourage open conversations about mental illness. Education must be tailored to the specific needs and experiences of African communities, considering the diverse cultural and religious backgrounds that influence attitudes toward mental health. Collaboration with community leaders, religious groups, and local organizations can also contribute to reducing stigma and promoting mental health. Creating a safe environment for dialog and providing resources for individuals and families to learn more about mental health can enable communities to address mental health issues collectively, leading to early intervention and better outcomes.

The findings of this study highlight the importance of using reliable, culturally appropriate tools, such as the CAMI scale, to evaluate attitudes toward mental illness in African communities. The CAMI scale is found to be a valuable tool for understanding the specific attitudes and beliefs of African Australians regarding mental health. By examining the existing barriers and negative perceptions, this study provides valuable insights for health care providers, policymakers, and community organizations to develop tailored interventions to improve attitudes toward mental illness. Ultimately, by adopting a collaborative approach to mental health awareness, education, and service delivery, we can build a more inclusive and compassionate society where individuals with mental illness are not only accepted but also encouraged to seek help without the fear of judgment or discrimination. This holistic approach is crucial in reducing mental health stigma and enhancing the overall mental health outcomes for African communities in Australia and beyond. We recognize that our conclusions regarding the relevance of the CAMI scale may have been overstated. The small sample size (n = 110), low reliability scores for the Authoritarianism (α = 0.42) and Social Restrictiveness (α = 0.62) subscales, and significant non-response rates for key demographic variables indicate that the results should be considered preliminary. The limited sample size also impacts the generalizability of the findings and raises concerns about potential sampling bias. While the CAMI scale shows promise, further validation with larger and more representative samples is necessary. We suggest that future research gather item-level data to facilitate a more thorough evaluation and refinement of the scale. Furthermore, this study provides initial evidence supporting the use of a culturally appropriate and psychometrically sound tool to assess attitudes toward mental illness in this population.

## Figures and Tables

**Table 1 healthcare-13-03115-t001:** Participants’ responses to the Authoritarianism component of CAMI scale.

	Strongly Agree	Agree	Don’t Know	Disagree	Strongly Disagree
One of the main causes of mental illness is a lack of self-discipline and will power	20(18.2%)	23(20.9%)	17(15.5%)	27(24.5%)	23(20.9%)
The best way to handle the mentally ill is to keep them behind locked doors	7(6.4%)	19(17.3%)	7(6.4%)	40(36.4%)	37(33.6%)
There is something about the mentally ill that makes it easy to tell them from normal people	38(34.5%)	42(38.2%)	11(10.0%)	14(12.7%)	5(4.5%)
As soon as person shows signs of mental disturbance, he should be hospitalized	36(32.7%)	40(36.4%)	12(10.9%)	11(10.0%)	11(10.0%)
Mental health patients need the same kind of control and discipline as a young child	32(29.1%)	42(38.2%)	9(8.2%)	19(17.3%)	8(7.3%)
Mental illness is an illness like any other	29(26.4%)	29(26.4%)	5(4.5%)	33(30.0%)	14(12.7%)
Less emphasis should be placed on protecting the public from the mentally ill	13(11.8%)	25(22.7%)	11(10.0%)	27(24.5%)	34(30.9%)
Mental hospitals are an outdated means of treating the mentally ill	19(17.3%)	20(18.2%)	11(10.0%)	31(28.2%)	29(26.4%)
Virtually anyone can become mentally ill	40(36.4%)	44(40.0%)	6(5.5%)	13(11.8%)	7(6.4%)

**Table 2 healthcare-13-03115-t002:** Participants’ responses to the Benevolence component of the CAMI scale.

	Strongly Agree	Agree	Don’t Know	Disagree	Strongly Disagree
The mentally ill have too long been the subject of ridicule	43(39.1%)	48(43.6%)	7(6.4%)	10(9.1%)	2(1.8%)
More tax money should be spent on the care and treatment of the mentally ill	44(40.0%)	50(45.5%)	9(8.2%)	3(2.7%)	4(3.6%)
We need to adopt a far more tolerant attitude towards the mentally ill in our society	48(43.6%)	50(45.5%)	3(2.7%)	7(6.4%)	2(1.8%)
Our mental hospitals seem more like prisons than like places where the mentally ill can be cared for	42(38.2%)	42(38.2%)	10(9.1%)	12(10.9%)	4(3.6%)
We have a responsibility to provide the best possible care for the mentally ill	57(51.8%)	39(35.5%)	2(1.8%)	8(7.3%)	4(3.6%)
The mentally ill don’t deserve our sympathy	12(10.9%)	5(4.5%)	3(2.7%)	30(27.3%)	60(54.5%)
The mentally ill are a burden on society	12(10.9%)	21(19.1%)	11(10.0%)	30(27.3%)	36(32.7%)
Increased spending on mental health services is a waste of taxpayers’ money	6(5.5%)	10(9.1%)	4(3.6%)	38(34.5%)	52(47.3%)
It is best to avoid anyone who has mental problems	14(12.7%)	11(10.0%)	10(9.1%)	35(31.8%)	40(36.4%)

**Table 3 healthcare-13-03115-t003:** Participants’ responses to the Social Restrictiveness component of CAMI scale.

	Strongly Agree	Agree	Don’t Know	Disagree	Strongly Disagree
The mentally ill should not be given any responsibility	19(17.3%)	28(25.5%)	14(12.7%)	34(30.9%)	15(13.6%)
The mentally ill should be isolated from the rest of the community	14(12.7%)	19(17.3%)	10(9.1%)	32(29.1%)	35(31.8%)
A person would be foolish to marry someone who has suffered from mental illness, even though he/she seems fully recovered	11(10.0%)	14(12.7%)	15(13.6%)	43(39.1%)	27(24.5%)
I would not want to live next door to someone who has been mentally ill	11(10.0%)	23(20.9%)	13(11.8%)	42(38.2%)	21(19.1%)
Anyone with a history of mental problems should be excluded from taking public office	16(14.5%)	23(10.9%)	13(11.8%)	30(27.3%)	28(25.5%)
The mentally ill should not be denied their individual rights	47(42.7%)	44(40.0%)	4(3.6%)	9(8.2%)	6(5.5%)
Mental health patients should be encouraged to assume the responsibilities of normal life	32(29.1%)	47(42.7%)	8(7.3%)	17(15.5%)	6(5.5%)
No one has the right to exclude the mentally ill from their neighborhood	29(26.4%)	46(41.8%)	9(8.2%)	14(12.7%)	12(10.9%)
The mentally ill are far less a danger than most people suppose	24(21.8%)	28(25.5%)	20(18.2%)	26(23.6%)	12(10.9%)
Most women who were one patient in a mental hospital can be trusted as babysitters	12(10.9%)	19(17.3%)	19(17.3%)	32(29.1%)	28(25.5%)

**Table 4 healthcare-13-03115-t004:** Participants’ responses to the Community Mental Health Ideology.

	Strongly Agree	Agree	Don’t Know	Disagree	Strongly Disagree
Residents should accept the location of mental health facilities in their neighborhood	25(22.7%)	50(45.5%)	14(12.7%)	14(12.7%)	7(6.4%)
The best therapy for many mental health patients is to be part of a normal community	22(20.0%)	51(46.4%)	12(10.9%)	19(17.3%)	6(5.5%)
As far as possible, mental health services should be provided through community-based facilities	35(31.8%)	54(49.1%)	7(6.4%)	9(8.2%)	5(4.5%)
Locating mental health services in residential neighborhood does not endanger local residents	16(14.5%)	46(41.8%)	14(12.7%)	22(20.0%)	12(10.9%)
Residents have nothing to fear from people coming into their neighborhood to obtain mental health services	16(14.5%)	60(54.5%)	12(10.9%)	17(15.5%)	5(4.5%)
Mental health facilities should be kept out of residential neighborhood	17(15.5%)	32(19.1%)	11(10.0%)	36(32.7%)	14(12.7%)
Local residents have good reason to resist the location of mental health services in their neighborhood	18(16.4%)	38(34.5%)	11(10.0%)	29(26.4%)	14(12.7%)
Having mental health patients live within residential neighborhood might be good therapy but the risks for the residents are too great	23(20.9%)	49(44.5%)	11(10.0%)	19(17.3%)	8(7.3%)
It is frightening to think of people with mental health problems living in residential neighborhoods	16(14.5%)	40(36.4%)	17(15.5%)	30(27.3%)	7(6.4%)
Locating mental health facilities in a residential area downgrades the neighborhood	9(8.2%)	19(17.3%)	14(12.7%)	38(34.5%)	30(27.3%)

**Table 5 healthcare-13-03115-t005:** Presentation of the Reliability of the scales.

	Cronbach’s Alpha	Split-Half Reliability Coefficient			Correlation Between Form
		Alpha 1	Alpha 2	Spearman–Brown	
Authoritarianism	0.424	0.478	0.431	0.068	0.04
Benevolence	0.730	0.527	0.780	0.529	0.36
Social restrictiveness	0.627	0.772	0.434	0.171	0.09
Ideology	0.724	0.709	0.756	0.306	0.18
Community attitude towards mental illness	0.717	0.567	0.625	0.609	

## Data Availability

The data presented in this study are available on request from the corresponding author. The data are not publicly available due to privacy or ethical restrictions.

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
