# Peer review of "Exploring African Community Attitudes Towards Mental Illness in Australia: A Cross-Sectional Study"

_healthcare, 2025, doi:10.3390/healthcare13233115_

Round 1
Reviewer 1 Report
Comments and Suggestions for Authors
The manuscript entitled “Exploring African Community Attitudes Towards Mental 2 Illness in Australia: A Cross-Sectional Study” proposes to evaluate the psychometric properties of the CAMI scale within African communities in Australia.
Although the premise is interesting, I believe the paper has several points that make it unsuitable for publication in its current form.
I'll highlight some of these points below:
1. The premise of the study is to evaluate the psychometric properties of the CAMI. However, only the internal consistency of the instrument and the associations between its domains were assessed. Other properties should also be examined, such as its repeatability (test-retest reliability), discriminant validity (if applicable), etc.
2. In the Sample Size section, it is stated that the ideal sample would be between 200 and 400 participants, but the study used a sample of 110 valid responses. Is this sample size sufficient for a 40-item instrument? Given the small sample size, the results presented for the alpha and correlation coefficients should be reported with their respective confidence intervals.
3. Looking at Table 1, one can observe a considerable number of non-responses in the study. Did the same rate of non-responses occur with the CAMI items as well? In lines 193-194 of the first version of the manuscript, it says: "... no participants had missing values beyond imputation by the half-rule." What was the amount of missing values that were imputed? This information should be included in the manuscript.
4. The authors use the CAMI, which consists of 40 items divided into 4 domains. However, when checking Tables 2 to 5, it appears that only 38 items are listed. Where are the two missing items?
5. Table 6: How should the values of the “Split-Half Reliability Coefficient” results be interpreted? Unlike Cronbach’s alpha, the authors did not indicate what thresholds are considered adequate. Furthermore, it is also necessary to clarify the role of the Spearman-Brown coefficient (how should this coefficient be interpreted, and what does the result observed in this study imply?).
Reviewer 2 Report
Comments and Suggestions for Authors
Does the introduction provide sufficient background and include all relevant references?
The introduction provides a solid overview of global mental illness and stigma, as well as some context for African communities in Australia. However, the narrative is somewhat repetitive and could be more focused on the specific research gap that it addresses. Additional recent references on psychometric validation in culturally diverse groups would strengthen the rationale.
Is the research design appropriate?
A cross-sectional survey design is appropriate for the research question. However, the sample size (n=110) is considerably smaller than recommended for psychometric validation of a 40-item scale, which weakens the robustness of the findings. The use of convenience sampling also limits generalizability.
Are the methods adequately described?
The instrument and analytical procedures are described in detail. Still, the treatment of missing demographic data is not sufficiently addressed. High non-response rates for key variables (e.g., years in Australia, country of origin) raise concerns about bias and should be more explicitly acknowledged.
Are the results clearly presented?
Results are systematically reported with clear tables. However, some tables are too dense and could be simplified or moved to supplementary material. The low Cronbach’s alpha for Authoritarianism (0.42) and Social Restrictiveness (0.62) needs more cautious interpretation than currently presented.
Are the conclusions supported by the results?
The general conclusions about the applicability of the CAMI scale in this population are overstated. Given the small sample, low reliability in two subscales, and high demographic non-response, findings should be framed as preliminary. Strong claims of validation are not fully supported.
Are all figures and tables clear and well-presented?
Tables are comprehensive but visually heavy, which reduces readability. Figures are lacking—graphical summaries (e.g., distributions of attitudes) would make the results more accessible.
Required Corrections
- Reframe conclusions as preliminary findings rather than a strong validation of the CAMI scale.
- Provide a fuller discussion of sample size limitations and their impact on psychometric reliability.
- Acknowledge and analyze missing demographic data and potential bias more explicitly.
- Interpret subscale reliabilities with caution—avoid overstating the robustness of Authoritarianism and Social Restrictiveness.
- Simplify presentation: move extensive demographic tables to supplementary material and add graphical illustrations of main findings.
- Strengthen the introduction and discussion with more recent literature on psychometric validation in diverse populations.
- Streamline the text to reduce repetition and improve readability.
Final Recommendation: The study addresses a novel and important topic with the potential to inform culturally sensitive mental health practice. However, the methodological weaknesses—particularly the small sample size, missing demographic data, and low reliability of some subscales—substantially limit the strength of the conclusions. With significant revisions and reframing as exploratory research, the manuscript could make a valuable contribution to the literature.
Reviewer 3 Report
Comments and Suggestions for Authors
Dear Author,
The topic of studying the African community’s attitudes towards mental illness in Australia is of interest. I have a few comments for enhancement:
- Please add the study design to the abstract.
- The introduction is quite lengthy and focuses heavily on mental illness. I would appreciate it if you could focus more on the CAMI tool and its previous psychometric assessment.
- Please clarify the meaning of “African community attitudes toward mental illness and its relation in Australia,” as this point seems unclear.
- Highlight the significance of this study.
- Delete any redundancy in the introduction to make it more focused and concise.
- I suggest revising the references to include more recent sources (within the past five years).
- Kindly add the process of data collection
- Kindly clarify why there was no response regarding the demographic data in Table 1.
Good Luck,
Reviewer 4 Report
Comments and Suggestions for Authors
Here are my comments.
- The paper aims to describe the attitudes towards mental illness of African communities in Africa and determine its reliability. The motivation for this study is quite clear. The design of the study is appropriate for these purposes. However, the use of convenient sampling led to a very large concentration of particular groups (e.g. immigrants from West Africa, younger age and Christians) which may not be representative of the total population. The paper is well-written and comprehensive in presenting the results although there are significant gaps in the analysis of the data and problems with the Discussion.
- The authors seemed to treat the CAMI as a validated tool for African communities in Australia, even though validity was being tested in this study. It was used by the authors to describe attitudes towards mental illness as if it was valid. This being so, the authors should have tested the CAMI for validity in African communities in Australia, instead of reliability only.
- The authors apparently conclude that CAMI is a valid tool ("The study examined the psychometric reliability of the CAMI scale for African migrant populations. ....... This research not only validates the CAMI scale for use with African migrants in Australia but also contributes to the broader field of cross-cultural research in mental health."). I suggest that the authors refrain from claiming that their study demonstrated the validity of CAMI in African communities in Australia because this was not really done. Reliability does not guarantee validity.
- The testing for reliability was inadequate, using only Cronbach's alpha and split-half reliability. Even for these, there are some gaps in this determination. The authors should have identified which items within subdomains (authoritarianism, benevolence, social restrictiveness, etc.) do not relate to its subdomain and that cause very poor consistency, such as the case for authoritarianism and social restrictiveness. These unrelated items might have to be removed from the tool as these do not measure the purported subdomain. It would have been nice if the authors showed the results of the correlation of the score of each item with the total score in its corresponding domain for the reader to be able to identify problematic items.
- Over-all Cronbach's alpha of 0.717 is just above the borderline acceptable value of 0.70. For a 40-item tool, this might not mean that much since Cronbach's alpha is known to correlate with the number of items. The Cronbach's alpha is related to the average of the correlations. There could be many correlations that are significant but also many possible very low correlations.
- The authors did not discuss what are the implications of split-half reliability coefficients that are not consistent, such as those for benevolence (0.527 and 0.780) and social restrictiveness (0.772 and 0.434). This seems to be indicative of the presence of items that are not related in its domain.
- The authors said that "for all analyses, the scores were re-aranged so that higher values correspond to more favorable attitudes towards mental health". How then can the authors explain the statistically significant negative correlation between benevolence and social restrictiveness and between social restictiveness and mental health ideology? The authors were conspicuously silent about this result.
- In the Discussion, there was no mention about the implications of the reliability coefficient of each domain in determining attitudes towards mental illness of African communities in Australia. The results should have also led to the identification of possible improvements of the measurement in order to improve its reliability. This would be a good contribution to knowledge.
- Some statements of the authors in the Discussion are not supported by the results of the study. Examples of these are 1) "In this study, almost three quarters of the participants, particularly those with authoritarianism idea, still held a stigmatised view of mental illness." and 2) "Majority of the participants with benevolence view in the CAMI scale provided their thoughts on the need for financial investment in mental health services. More than 80% of the participants with this view argued that every investment in this area is money well-spent." These are not presented in the results. The authors could have done an analysis of these given their data to support these statements.
- The title of the tables are not very informative. This is only one word that describes them.
- The occurrence of large missing values for some variables (country of origin and number of years lived in Australia) is noticeable and their implications cannot be ignored.
- There are studies cited in the text do not appear in the reference and conversely, there are references that were not cited in the text.
Round 2
Reviewer 1 Report
Comments and Suggestions for Authors
I understand the study’s limitations regarding the small sample size, and I believe the authors have adequately addressed the issues raised and provided justifications/clarifications in the manuscript in a satisfactory manner.
Author Response
We thank the reviewer for acknowledging the study’s limitations and for recognizing the clarifications and justifications we provided regarding the small sample size. We appreciate your positive assessment and are grateful that you found our revisions satisfactory.
Reviewer 2 Report
Comments and Suggestions for Authors
The authors have adequately addressed all the reviewers’ comments and implemented the necessary revisions in a clear and comprehensive manner. The manuscript has significantly improved in clarity, structure, and scientific rigor. The methodological explanations are now more transparent, the interpretation of findings is balanced, and the limitations are appropriately acknowledged. Overall, I believe the paper is now of publishable quality and can be accepted for publication in its current form.
Author Response
We sincerely thank the reviewer for the positive and encouraging feedback. We appreciate your recognition of the revisions made to improve the clarity, structure, methodological transparency, and scientific rigor of the manuscript. We are grateful for your conclusion that the paper is now of publishable quality.
Reviewer 4 Report
Comments and Suggestions for Authors
In my previous review, I suggested that for those domains that have low Cronbach's alpha (e.g. authoritarianism), some of the items in these domains that do not correlate well could be identified. These could be done using the item-to-total correlations and pairwise correlations of the items in the domain. This could have been done in this study. Instead, the authors recommended that this be done in a future study.
This is just a suggestion. My approval for publication does not require this to be done.
Author Response
We thank the reviewer for the helpful suggestion regarding examining item-to-total and pairwise correlations for domains with lower Cronbach’s alpha. We acknowledge the value of this additional analysis. As noted, we did not include it in the current study because it fell outside the planned scope, but we agree it would be a meaningful direction for future research. We also appreciate the reviewer’s clarification that this suggestion is not required for publication.